# Definition of the Pnictogen Bond: A Perspective

**Arpita Varadwaj [1,\*], Pradeep R. Varadwaj [1,2,\*], Helder M. Marques [2] and Koichi Yamashita [1]**

[1] Department of Chemical System Engineering, School of Engineering, The University of Tokyo 7-3-1, Tokyo 113-8656, Japan

[2] Molecular Sciences Institute, School of Chemistry, University of the Witwatersrand, Johannesburg 2050, South Africa

\* Correspondence: varadwaj.arpita@gmail.com (A.V.); pradeep@t.okayama-u.ac.jp (P.R.V.)

**Abstract:** This article proposes a definition for the term "pnictogen bond" and lists its donors, acceptors, and characteristic features. These may be invoked to identify this specific subset of the inter- and intramolecular interactions formed by elements of Group 15 which possess an electrophilic site in a molecular entity.

**Keywords:** pnictogen bond; nomenclature; non-covalent interactions; molecular interactions; crystallography; self-assembly; supramolecular chemistry; catalysis; photovoltaics; nanomaterials

## 1. Preface

This paper proposes a definition of the term "pnictogen bond", followed by a list of electron density donors and acceptors of pnictogen bonds and their accompanying experimental and theoretical features. It proposes that the definition be used to designate a subset of the family of inter- and intramolecular interactions formed by the members of the pnictogen family [1], the elements of Group 15 of the periodic table, in molecular entities. This proposal follows from the IUPAC recommendations for hydrogen bonds (HBs) [2], halogen bonds (XBs) [3], and chalcogen bonds (ChBs) [4].

Nitrogen, the lightest member of the pnictogen family, Group 15, has the highest electronegativity and lowest polarizability [5]. It often serves as a nucleophile when present in molecules, such as in $N_2$, $NH_3$, and ammine derivatives, for example [6–10]. The heavier members of the family exhibit similar behavior when they are a constituent of many chemical systems. This presumably applies to Moscovium as well, although little is known about its chemistry. As electron density donors, they are capable of acting as acceptors of, inter alia, hydrogen bonds, halogen bonds, chalcogen bonds, and any other non-covalent interaction. In such cases, the pnictogen atom behaves as a nucleophilic moiety that attractively engages (via coulombic interaction) with its interacting electrophilic partner(s).

The pnictogen atoms in molecular entities also have the ability to act as electron-poor (electrophilic) sites [6,7,11–14]. This occurs when they are bonded to electronegative and/or electron-withdrawing groups such as F, CN, $NO_2$, and $C_6F_5$. In such instances, they are capable of attracting an electron-rich (nucleophilic) site in the same or in a separate molecular entity when in close proximity.

The difference between the two situations described above depends on the electronic structure profile of the bound pnictogen atom in the molecular entity; it acts as a nucleophile in the first case and as an electrophile in the second. In the latter case, the bound pnictogen atom may be directionally oriented toward the nucleophilic site, resulting in the development of a linear or quasi-linear non-covalent interaction [6,7,11–14]. If the entire electrostatic surface of the bound pnictogen atom in a molecular entity is electrophilic, it may lead to the formation of non-linear (or bent) attractive interactions [6,7,11–14]. The

term "pnictogen bond" is used uniquely to designate the latter set of non-covalent interactions, where the pnictogen atom acts as an electrophile. The presence of an electrophilic and a nucleophilic site on a pnictogen atom in molecular entities can be unequivocally identified by experimental and/or theoretical methods [6,7,11–14].

The IUPAC definition of hydrogen bonds was revised in 2011, adding various features, characteristics, and footnotes [2]. The same format was adopted for halogen bonds [3] and chalcogen bonds [4]; the only change in the definition was a change in the family name. Thus, in the definition of "halogen bond," "hydrogen" was replaced by "halogen"; the same was carried over to "chalcogen bond," with "hydrogen" being replaced by "chalcogen". The purpose of this was to unify the terminology of chemical bonding. The hydrogen, halogen, and chalcogen atoms in molecular entities that form hydrogen, halogen, and chalcogen bonds are electrophiles, i.e., they exhibit electrophilic properties while forming hydrogen, halogen, and chalcogen bonds in chemical systems, respectively.

In the near future, an IUPAC working group may recommend definitions for the terms "tetrel bond (TtB)", "pnictogen bond (PnB)", and any other non-covalent interaction. The first two terms have been increasingly used in the current literature to describe the attractive interactions formed when respective elements of Groups 14 and 15 act electrophilically on nucleophiles in the solid, liquid, and gas phases.

A brief definition of the term "pnictogen bond" is provided below, followed by illustrative examples in the form of a non-exhaustive list of common pnictogen bond donors and acceptors. This proposal was developed by reviewing the list of experimental and theoretical features already extensively documented in the literature. Although not comprehensive, we suggest that this definition and its accompanying features be used as potential signatures when attempts are made to identify and characterize pnictogen bonds in chemical systems.

## 2. Definition and Recommendations

A pnictogen bond occurs in chemical systems *when there is evidence of a net attractive interaction between an electrophilic region associated with a pnictogen atom in a molecular entity and a nucleophilic region in another, or the same molecular entity.*

Note 1: A pnictogen bond is usually represented by three dots in the geometric motif R–Pn⋯A, where Pn is the PnB donor, representing any pnictogen atom (possibly hypervalent) that has an electrophilic region on it; R is the remaining part of the molecular entity R–Pn containing the PnB donor; A is a PnB acceptor, which may or may not represent a molecular entity, but that has at least one nucleophilic region.

Note 2: An electrophilic site on the PnB donor Pn generally refers to the lowest electron density region, while a nucleophilic site on the PnB acceptor A usually refers to the highest electron density region, and the resulting interactions formed between the two entities exhibit different directional features and complementarity.

Note 3: At an equilibrium configuration, PnB donors Pn exhibit the ability to act as electron density acceptors, and PnB acceptors A exhibit the ability to act as electron density donors.

Note 4: A pnictogen bond may occur within a neutral molecule [12,14] or between two neutral molecules in close proximity [12,13]; it can also occur between a neutral molecule with a PnB donor Pn and an anion containing A [15]; between a PnB donor in a molecular cation and a nucleophile (or negative π-density) A on a neutral molecule [16]; between an electron-poor delocalized region (positive π-density) as the PnB donor Pn and nucleophile A (or negative π-density) on the acceptor entity; or between two molecular entities of opposite charge polarity (i.e., an ion pair)) with a PnB donor and a PnB acceptor [7,17,18].

Note 5: Because of its hypervalent character, a pnictogen atom in a molecular entity may form one or more than one pnictogen bond concurrently [6,11–14].

Note 6: Because of its variable electrostatic character, a pnictogen atom in a molecular entity may engage in a number of interactions that lead to the appearance of a variety electronic and geometric features [6,7,11–14,19]. The term pnictogen bond should not be used for attractive interactions in which the pnictogen atom (frequently nitrogen and sometimes phosphorous) functions as a nucleophile.

Note 7: The electrophilic and nucleophilic characteristics of a bound pnictogen atom and its PnB forming ability may be found by searching for the local minima and maxima of the potential on the electrostatic surface of the molecular entity [6,7,11–14,20–28]. The electrophilic region on the surface of the bound pnictogen atom along the outermost extension of the R–Pn covalent or coordinate bond in an isolated monomeric entity is often (but not always) represented by a local maximum of the potential and may be used to search for pnictogen bonds between it and the nucleophilic regions on atoms in the entities with which it interacts [6,7,11–14,20–28].

Note 8: Two pnictogen atoms in two different molecular entities may be involved in an attractive engagement to form a pnictogen bond, in which case, one of the pnictogen atoms must act as a pnictogen bond donor, and that in the partner molecular entity must act as a PnB acceptor, such as in $NO_2HP \cdots NH_3$ [29].

Note 9: The pnictogen bond should be viewed as an attractive interaction between PnB donor site Pn and PnB acceptor site A of opposite charge polarity ($Pn^{\delta+}$ and $A^{\delta-}$), resulting in a coulombic interaction between them; the charge polarity $\delta+$ and $\delta-$ symbolically refers to the local charge polarity on the interacting regions on Pn and A, respectively.

Note 10: The pnictogen bond should follow the Type-II topology of non-covalent bonding interactions; a Type-II interaction, R–Pn$\cdots$A, is often linear or quasi-linear (but may be non-linear) and satisfies Note 9.

### 3. Some Common Pnictogen Bond Donors and Acceptors

Some common PnB donors and acceptors are listed below. We emphasize that the list, which emerged from a search of the Cambridge Structural Database [30] and Inorganic Crystal Structure Database [31,32], is illustrative rather than comprehensive.

The PnB donor Pn can be:

–  A pnictogen in a trihalide: $PnX_3$ (Pn = N, P, As, Sb, Bi; X = halide).
–  Nitrogen in a geminal-difluoramine ($NF_2$) (as in *N,N*-difluoroamino-2,4-dinitrobenzene [33] and in 1-(3-(5,5-bis(difluoroamido)-2-oxopyrrolidinyl)-2-oxopropyl)pyrrolidine-2,5-dione, $C_{11}H_{12}F_4N_4O_4$ [34]).
–  The $N_\beta$ of covalently bonded azides, such as $-N_\alpha = N_\beta = N_\gamma$ (as in 2,4,6-triazidoborazine ($H_3B_3N_{12}$) [35], 5-diazonio-4-(2*H*-tetrazol-5-yl)-1,2,3-triazol-1-ide ($C_3HN_9$) [36]), and 2,2,4,4,6,6-hexaazido-2,4,6-triphospha-1,3,5-triazine ($P_3N_{21}$) [37], or nitrogen in the diazonio fragment, such as in $-N_\alpha = N_\beta$ (as in 4-diazonio-3,5-dinitropyrazol-1-ide ($C_3N_6O_4$) [38] and in diazonionaphthalen-1-olate ($C_{10}H_6N_2O$) [39].
–  The nitrogen in ammonium, diammonium, and (chain and arene) derivatives of ammonium (for example, $NH_4^+$, $NH_3NH_3^{2+}$, $NH_3NH_2^+$, $CH_3NH_3^+$, $[C_nH_{2n+1}NH_3]^+$ ($n$ = 2, 3, …, 18), and $[NH_3(CH_2)_mNH_3]^{2+}$ ($m$ = 2, 3, …, 8) [7]).
–  The pnictogen atom in many cations ($NH_3OH^+$ [40]; $C_6F_5ClP_5^+$; $AsMe_3H^+$; derivatives of $[Sb(C_6H_5)_4]^+$ (as in [16,18,41–44]; $[Sb(C_6H_5CH_3)_4]^+$ [42]; $[Bi(C_6H_5OCH_3)_3(CH_3)]^+$ [45]; $BiMe_4^+$; and $[Bi(C_6H_5)_4]^+$ [46–48] and its derivatives [49], etc.).
–  The nitrogen in a nitro group (e.g., $O_2NN(H)C(O)N_3$] [50] and $C_5H_5N_5O_3$) [51].

- The phosphorous in phosphoryl halides ($POF_3$, $POCl_3$, and $POBr_3$) [11]; phosphorus(V) triazides $OP(N_3)_3$ and $SP(N_3)_3$ [52]; diphosphorus tetraiodide $P_2I_4$, phosphorus tricyanide, $P(CN)_3$, and 4,4',4''-phosphinetriyltripyridine [53]; and disphospha-functionalised naphthalenes (such as $Nap(PCl_2)_2$ $Nap(PBr_2)_2$ and $Nap(PI)_2$ (Nap = naphthalene-1,8-diyl) [54]) and phosphorus diisocyanate chloride $P(CO)_2$ [55], etc.
- Phosphorous in derivatives of halo-substituted phosphazenes (viz. cyclotetrakis(difluorophosphazene) $F_8N_4P_4$ [56], decafluorocyclo-pentaphosphazene $F_{10}N_5P_5$ [57], hexachloro-cyclo-triphosphazene $Cl_6N_3P_3$ [58], cyclo-tetrakis(phosphorus(V) nitride dichloride) $Cl_8N_4P_4$ [59], nonachlorohexahydroheptaazahexaphosphaphenalene $Cl_9N_7P_6$, [60], tris(dibromophosphazene) $Br_6N_3P_3$ [61], octabromocyclotetraphosphazene $Br_8N_4P_4$, [62], etc.).
- Phosphorous in phosphorus oxides (phosphorus(V) oxide $P_2O_5$ [63], tetraphosphorus(III) oxide $P_4O_6$ [64], tetraphosphorus(III,IV) heptaoxide $P_4O_7$ [65], phosphorus(II) oxide $P_4O_8$ [66], tetraphosphorus(II,III) nonaoxide oxide $P_4O_9$ [67], tetraphosphorus(V) oxide $P_4O_{10}$ [68], phosphorus ozonide $P_4O_{18}$ [69], etc.).
- Pnictogen in halide-, amino-, imidazole-, oxy-, and thio-substituted heavier pnictogen derivatives, in diaryl halido-substituted bismuthanes (e.g., $C_{24}H_{34}BiI$ [70]), and in $BiMe_3Cl_2$, $AsMe_3$, $SbMe_3$, $BiMe_3$, etc.).
- The arsenic atom in methylenebis(dichloroarsane) [71], 5,10-epithio-5,10-dihydroarsanthrene [72], 8,8'-(phenylarsanediyl)diquinoline [73], 2-chloro-1,3-dimethyl-1,3-diaza-2-arsolidine [74], 4-(1,3,2-dithiarsinan-2-yl)aniline [75], etc.
- The antimony in bis(dimethylstibanyl)sulfane [76], bis(dimethylstibanyl)oxane [76], (trimethyl-stibino)-dimethyl-stibonium [77], trichloro-dipyridine-antimony [78], triphenyl-bis(p-tolylacetato)-antimony [79], bis(3-methoxyphenylacetate)-triphenylantimony [79], bis(acetato-O)-(2,6-bis(t-butoxymethyl)phenyl-C)-antimony(III) [80], bis(trichloro-antimony) [81], etc.
- Arene-substituted pnictogen derivatives, including the bismuth in triphenylbismuth $Bi(C_6H_5)_3$ and pyridine dipyrrolide complexes, $C_{43}H_{37}BiIN_3$, etc.
- A positive $\pi$ system (species featuring a double or triple bond (e.g., midpoint of the $N{\equiv}N$ bond in $N_2$; P in $P_2$; Bi in $Bi_2$; N in $NO_2$) of neutral and cationic entities).

The PnB acceptor entity A can be:

- A lone pair on an atom in a molecule. There are almost limitless possibilities, for example, the N in pyridines or amines, or even in $N_2$; the O in $H_2O$, CO, $CO_2$, an ether, or a carbonyl group, or a phosphorus oxide; covalently bonded halogens in molecules; As in $AsMe_3$; a chalcogen in a heterocycle such as a thio-, seleno-, and tellurophene derivatives as well as fused polycyclic derivatives thereof; furoxans, 2,5-thiadiazoles N-oxides, sulfoxide, aryl sulfoxides, and tellurazoles N-oxides; derivatives of macrocyclic crown-ethers such as 18-crown-6, 15-crown-5 and 21-crown-7, etc.
- Many anions, such as halide anions; $NO_3^-$; $CF_3SO_3^-$; $BF_4^-$; tetraphenylborate $C_{24}H_{20}B^-$; $ClO_4^-$; 5-oxotetrazole $CHN_4O^-$; $I_3^-$; $Br_3^-$; $N_3^-$; $BF_4^-$; $AuCl_4^-$; $PF_6^-$; $AsF_6^-$; pentazolide $N_5^-$; 5,5'-bistetrazolates $C_2N_8^{2-}$; $p$-tosylate $C_7H_7SO_3^-$; polyatomic oxyanions such as $C_2O_4^{2-}$; $GaCl_4^-$; $ZnCl_4^-$; $ReO_4^-$; $AsCl_4^-$, $SbCl_4^-$; $BiCl_4^-$; etc.
- A (negative) $\pi$ system (species featuring a double or triple bond) and arene moieties of any kind, such as the centroid of the arenes and the midpoints of molecular $As_2$ and N in $NO_3^-$, etc.

## 4. Examples of Chemical Systems Featuring Pnictogen Bonding

The attractive intermolecular interactions between the nucleophilic regions on the nitrogen atoms in $N_2$, ammonia, alkyl cyanides, or amine derivatives and the electrophilic protons of other amine, halogen, and chalcogen derivatives are not pnictogen bonds; they are hydrogen bonds, halogen bonds, and chalcogen bonds, respectively (see Figure 1a–c, respectively, which show the N in methyl cyanide as the nucleophile). The attractive intermolecular interaction between Bi in a bismuth trihalide and N in amine derivatives is

a pnictogen bond. The attractive interaction between As, Sb, or Bi containing chemical systems (not all, but many) and O in water, carboxylic acids, aldehydes, nitrates, and ketones, etc., is a pnictogen bond, not a chalcogen bond (for example, the As···O pnictogen bond between the PnB donor atom As and the acceptor atom O in 2-(dicyanoarsino)-1,3-diisopropyl-4,5-dimethyl-1*H*-imidazol-3-ium [82]; the Sb···O pnictogen bond between the PnB donor atom Sb and the PnB acceptor atom O (acetato-O in a pair of two interacting building blocks in crystalline (R)-tris(trifluoroacetato-O)-antimony(III); the Sb···O pnictogen bond between Sb and the O in crystalline 8-(diphenylphosphino)naphthalen-1-yl)-triphenyl-antimony trifluoromethanesulfonate (CSD ref code: APOXIO) [41]; and the Sb···O pnictogen bond in 1,4-bis(2-nitrophenyl)-1,4-diarsa-2,3,5-trithiacyclopentane) (CSD ref code: ASADUT) [83]. The attractive intermolecular interaction between the Bi or Sb of bismuth or antimony trihalides and the O sites in crown ether derivatives is a pnictogen bond [84–88]. The attractive intramolecular interaction between antimony and chlorine in crystalline dichloro-triphenyl-antimony-trichloro-antimony (CSD ref: BUMGEV) [89] is not a halogen bond, it is a pnictogen bond. The attractive intramolecular interaction between arsenic and bromine in crystalline dibromo-trimethyl-arsenic is not a halogen bond, it is pnictogen bond [90]. Additionally, the attractive intramolecular interaction between arsenic and chlorine in crystalline 3,4,5,6-tetrachloro-1,2-diarsa-closo-hexaborane is a pnictogen bond, not a halogen bond [91]. The pnictogen-centered Pn···A intermolecular interactions between interacting units in selected crystalline solids shown in Figure 1d–i, viz. P···F (Figure 1d), P···O (Figure 1e,f), As···F (Figure 1g), Sb···S (Figure 1h), and Bi···π (Figure 1i), are pnictogen bonds and are not chalcogen bonds, halogen bonds, or tetrel bonds.

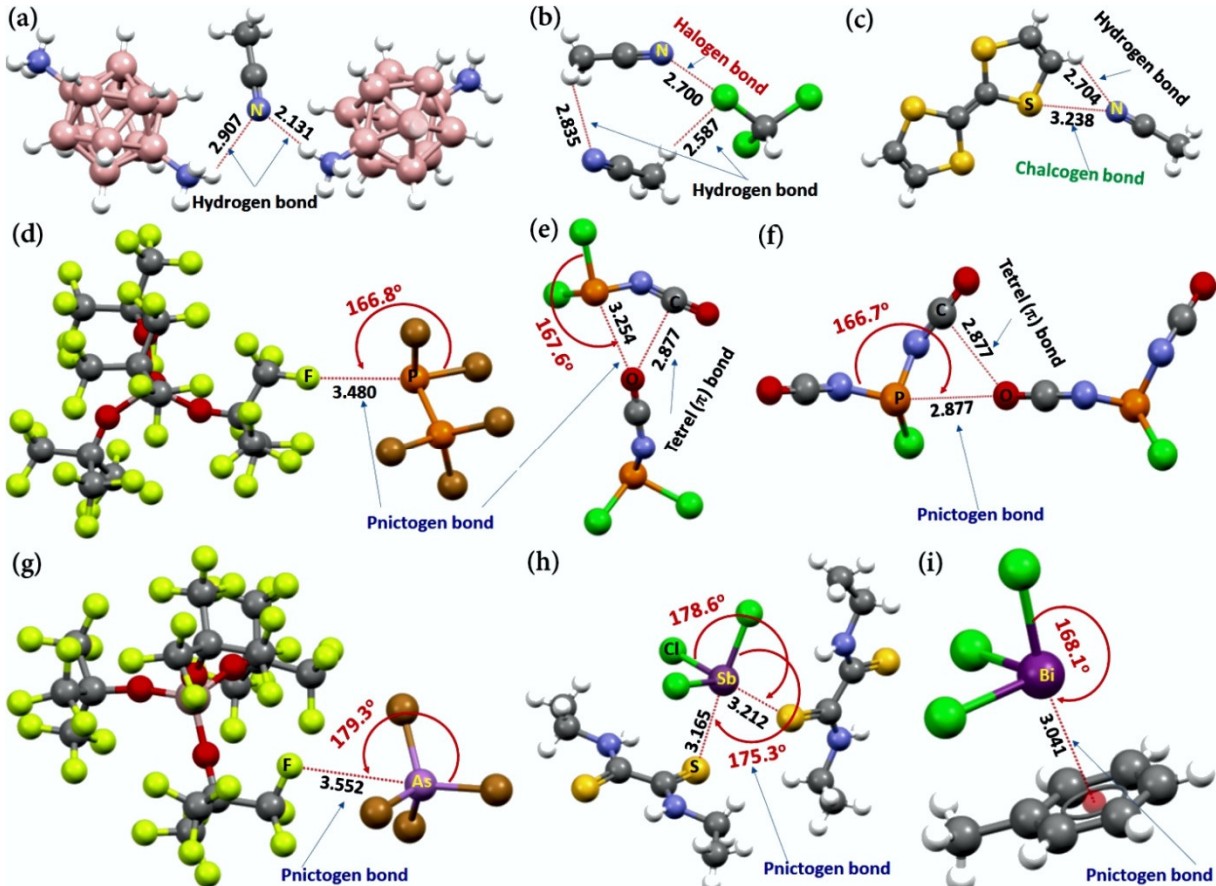

**Figure 1.** Illustration of various types of non-covalent interactions between building blocks found in selected crystalline materials: (**a**) 1,12-diammonia-closo-dodecaborate acetonitrile solvate (CSD ref: HESCAM) [92]; (**b**) bis(μ4-*h*2-vinylidene)-bis(μ2-bis(diphenylphosphino)methane)-dodecacar-

bonyl-hexa-ruthenium (CSD ref: AGOQOC) [93]; (**c**) tetrathiafulvalene 9-dicyanomethylene-2,7-di-nitrofluorene (CSD ref: AHAWUD) [94]; (**d**) pentabromodiphosphonium tetrakis(perfluoro-t-butoxy)-aluminium (HUHJID) [95]; (**e**) phosphorisocyanatidous dichloride (CSD ref: ITOLIO) [96]; (**f**) phosphorodi-isocyanatidous chloride (CSD ref: ALOYOS) [55]; (**g**) tetrabromoarsonium tetrakis(tris(trifluoromethyl)methoxy)-aluminium (CSD ref: XALVOW) [97]; (**h**) sesqui(*N*,*N'*-di-ethyldithio-oxamide) trichloro-antimony (CSD ref: BODBOL) [98]; and (**i**) trichloro-bismuth toluene (CSD ref: WIKDIE) [99]. Selected intermolecular bond angles and bond lengths are given in Å and degrees, respectively. Dotted lines between PnB donor atom Pn and PnB acceptor atom A represent an attractive interaction; the same is true for other noncovalent interactions, such as the hydrogen bond in (**a**–**c**) and tetrel (π) bond in (**e**–**f**).

Some further examples of chemical systems illustrating the variable nature of the geometric appearance of pnictogen bonding between the various PnB donor atoms Pn and various PnB acceptor atoms A in the interacting molecular entities are shown in Figure 2 (Note 3). The PnB donor atoms Pn are either in neutral species (Figure 2a–d,j–k), or in cations (Figure 1e–i), or in π-systems (Figure 2m–p); the PnB acceptor sites A are either lone pairs in neutral molecules (Figure 2a–e,m–p), in anions (Figure 2f–h,j), or π-systems (Figure 2i,k,l).

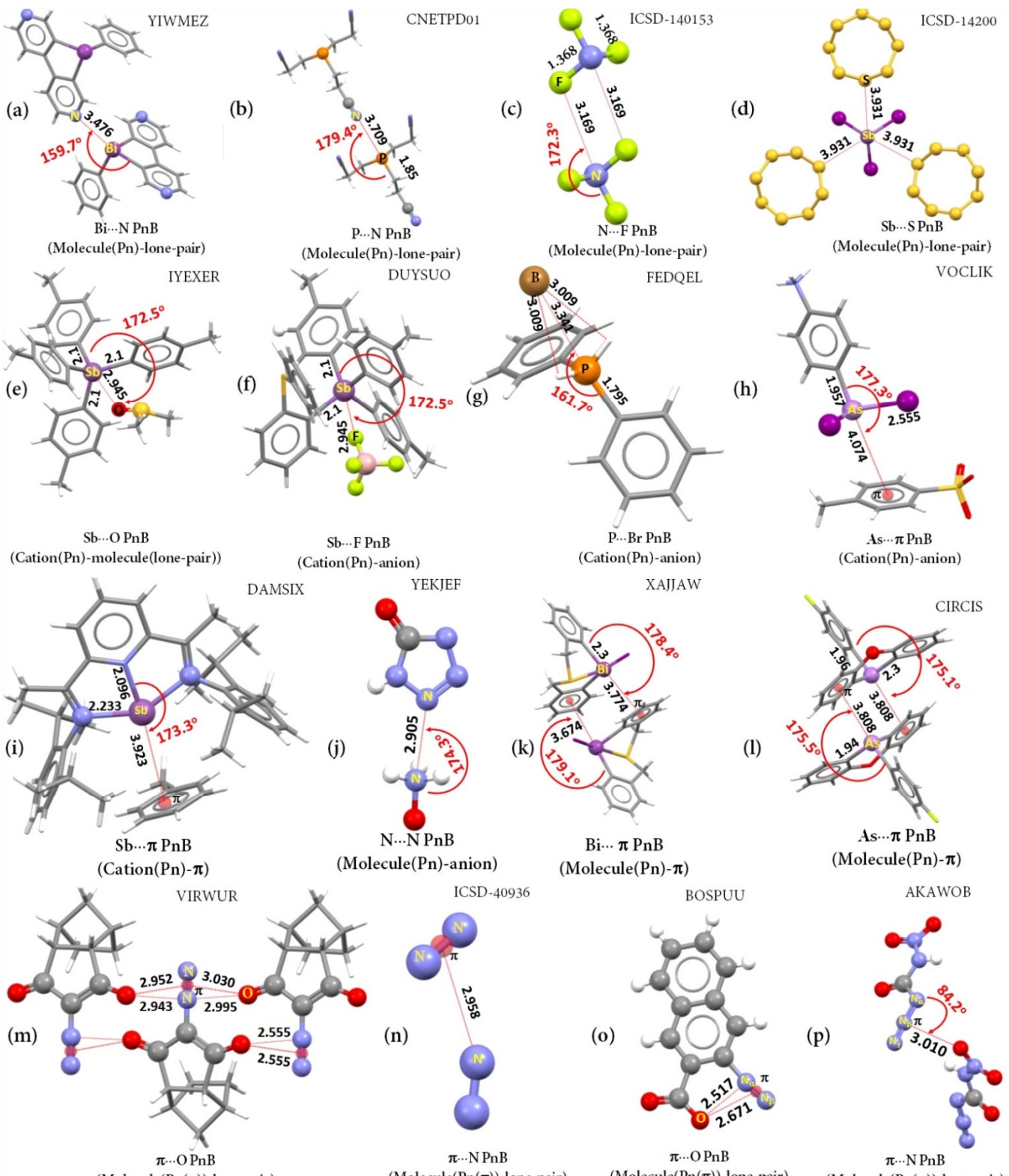

**Figure 2.** Some selected solid-state systems featuring pnictogen bonding: (**a**) 9-phenyl-9H-bismolo [2,3-c:5,4-c']dipyridine [100]; (**b**) 3',3'',3'''-Phosphinetriyltripropionitrile [101]; (**c**) trifluoroamine [102]; (**d**) antimony triiodide molecular sulfur S8 [103]; (**e**) bis(tetrakis(4-methylphenyl)-antimony) hexabromo-iridium [104]; (**f**) tris(4-methylphenyl)-(2-(phenylsulfanyl)phenyl)-antimony tetrafluoroborate [42]; (**g**) diphenylphosphenium bromide [105]; (**h**) 4-(di-iodoarsino)anilinium 4-methylbenzenesulfonate [106]; (**i**); (1,1'-(pyridine-2,6-diyl)bis[N-[2,6-di-isopropylphenyl]ethan-1-imine])-antimony [107] (**j**) 5-oxo-4,5-dihydrotetrazol-1-ide azane oxide [40]; (**k**) (2,2'-[sul-

fanediylbis(methylene)]di(phenyl))-iodo-bismuth(III) [108]; (**l**) 10-(4-fluorophenyl)-10H-phenoxar-sinine [109]; (**m**) 2-diazonio-1-oxo-3a,4,5,6,7,7a-hexahydro-1H-4,7-methanoinden-3-olate [110]; (**n**); molecular nitrogen [111]; (**o**) naphthalene-2-diazonium-3-carboxylate [112]; and (**p**) azido-nitra-mino-carbonyl [50]. Selected bond lengths and bond angles are in Å and degrees, respectively. Cambridge Structural Database [30] references are in uppercase letters, and Inorganic Crystal Structure Database [31,32] references are in numbers. Selected atoms acting as PnB donor and PnB acceptor in (**a**–**p**) are marked. Dotted lines between PnB donor atom Pn and PnB acceptor atom A represent an attractive interaction.

The geometric arrangement between the interacting entities in Figure 2a–d shows how pnictogen bonds are formed between the electrophilic sites on Bi, P, N, and Sb and a lone pair on N (in $C_{16}H_{11}BiN_2$), N (in $(P(CH_2)_2CN)_3$), F (in $NF_3$), and S (in $S_8$), respectively. Shown in Figure 2e–I are the PnB donor atoms Sb, Sb, N, As, and Sb, which sustain attractive interactions with the PnB acceptor sites N (lone pair on N in neutral $(CH_3)_2SO$), F (lone pair on F in anionic $BF_4^-$), Br (lone pair in Br in $Br^-$), and $\pi$ (arene moiety in the anion $[C_6H_4(CH_3)(SO_3)]^-$) and $\pi$ (arene moiety in toluene), respectively. Similarly, the geometric arrangement between the interacting entities in Figure 2j–l represents PnB donor atoms N, Bi, and Si in neutral entities that sustain attractive interactions with the PnB acceptor sites N and $\pi$ in interacting systems. In the systems in Figure 2m–p, the PnB donor site $N(\pi)$ of an $R-N_2$ or $R-N_3$ entity are attractively engaged with the nucleophiles on the O or N sites of the PnB acceptors.

## 5. A List of Characteristic Features

Evidence of the presence of a pnictogen bond in molecular entities, crystals, and nano-scale materials may emerge from experimental measurements (e.g., X-ray diffraction, infrared, Raman and NMR spectroscopy, etc.), or signatures from ab initio studies, or a combination of both. The evidence could be very similar to that already recommended by the IUPAC for HBs, HBs, and CBs. The following list is not exhaustive but includes some distinguishing features that may be useful as indicators of the occurrence of pnictogen bonding interactions in chemical systems. The more of these features that are met, the more reliable is the identification of the interaction as being a PnB interaction.

On the formation of a typical pnictogen bond R–Pn···A between two interacting entities:

a.   The separation distance between the PnB donor atom Pn and the nucleophilic site of PnB acceptor A tends to be smaller than the sum of the van der Waals radii of the respective interacting atomic basins [6,7,11–14] and larger than the sum of their co-valent bond radii [2–4]; the deviation of the former is likely since the known van der Waals radii of atoms are only accurate with ±0.2 Å [13,113,114];

b.   The PnB donor site on Pn tends to approach the PnB acceptor site A along the outer extension of a σ covalent or coordinate bond, and the angular deviation from the extension is often more pronounced in PnBs [6,7,11–14] than in halogen bonds, as in ChBs [4], with the latter possibly being due to the involvement of secondary interactions;

c.   The angle of interaction, ∠R–Pn···A, tends to be linear or quasi-linear when the approach of the electrophile on Pn is along the σ covalent/coordinate bond extension, but this can be non-linear or have a bent shape when the pnictogen bond occurs between an electron density-deficient (electrophilic) π-type orbital of the bonded pnictogen atom and the nucleophilic region on A [6,7,11–14] or when secondary interactions are involved;

d.   When the nucleophilic region on the PnB acceptor site A is a lone pair orbital or an electron density-rich π region, the PnB donor Pn tends to approach A along the axis of the lone pair or orthogonally to the π bond plane [6,7,11–13,115];

e.   The distance of the R–Pn covalent bond opposite to the PnB in a molecular adduct is typically longer than that in the isolated (unbound) PnB donor;

f.　The infrared absorption and Raman scattering observables of both R–Pn and A are affected by PnB formation; the vibrational frequency of the R–Pn bond may be red-shifted or blue-shifted depending on the extent of the interactions involved compared to the frequency of the same bond in the isolated molecular entity; new vibrational modes associated with the formation of the Pn⋯A intermolecular pnictogen bond should also be characteristically observed [116,117], as observed for ChBs;

g.　A bond path and a bond-critical point between Pn and A may be found when an electron density topology analysis based on the quantum theory of atoms in molecules (QTAIM) [118] is carried out, together with the emergence of other charge density-based signatures [119–123];

h.　Isosurface volumes (colored greenish, blue, or mixed blue–green between Pn and A, representative of attractive interactions [6,7,11–13,124]) may be seen if a non-covalent index analysis based on reduced charge density gradient [125–127] is performed;

i.　The UV–vis absorption bands of the PnB donor chromophore may experience a shift to longer wavelengths [128];

j.　At least some transfer of charge density from the frontier PnB acceptor orbital to the frontier PnB donor orbital may occur [15,129,130]; when the transfer of electron charge density between them is significant, the formation of a dative coordinate interaction is likely [131]; the occurrence of the IUPAC-recommended phenomena for HBs (see Criteria E1 and Characteristic C5 of Ref. [2]) is also applicable to XBs [132–135] and ChBs [136–138];

k.　The NMR chemical shifts of the nuclei in both R–Pn and A [4,128,139–143] are typically affected, as found for R–X⋯A XBs and R–Ch⋯A ChBs [144];

l.　The PnB strength typically decreases with a given acceptor A, as the electronegativity of Pn increases in the order Bi < Sb < As < P < N, and the electron withdrawing ability of R decreases;

m.　The PnB bond strength increases for a specific PnB acceptor A and the remaining R, as the polarizability of the pnictogen atoms in the molecular entities increases (Bi > Sb > As > P > N) [15]. This is the same as what is observed for the halogen derivative forming XB (I > Br > Cl > F) [145,146] and the chalcogen derivatives forming ChB (Te > Se > S > O) [147]. However, if secondary interactions (e.g., a hydrogen bond, halogen bond, chalcogen bond, tetrel bond, etc.) are simultaneously involved with either the PnB donor or PnB acceptor, the order of interaction strength may also be altered;

n.　Coulombic interaction occurs between the PnB donor and the PnB acceptor entities at equilibrium, and the energetic contributions to the binding energy arising from electrostatic polarization (and/or induction), exchange repulsion, and long-range dispersion should not be neglected [15,148].

## 6. Concluding Remarks

Pnictogen bonding is a non-covalent interaction with the potential to serve as an electronic glue in the assembly of molecular entities in the process of developing molecular complexes, crystalline solids, supramolecular structures, and functional nanomaterials. Its implications in crystal engineering [6,7,11–14,123,149–151], anion transport [152], catalysis [20,153–155], and photovoltaics [7,156,157] are appreciable. A pnictogen bond falls under the umbrella of σ- and/or π-hole-centered non-covalent interactions, provided that it is a result of an attractive engagement between an electrophilic PnB donor moiety Pn containing a σ- and/or a π-hole interacting with a nucleophilic site on A [6,7,11–14,158–162]; σ- and π-holes are electron density-deficient electrophilic regions on the PnB donor moiety Pn along the outermost extension of the R–Pn covalent (or coordinate) bond and perpendicular to that bond, respectively. The list of PnB donors and PnB acceptors and the characteristic features of PnB is vast, and only a few are listed in this paper. Several illustrative examples are provided that can assist in recognizing chemical situations where pnictogen bonding in and between molecular entities is likely to occur. The definitions

and features proposed in this paper should be useful for researchers and graduate students working in diverse research fields to identify and characterize pnictogen bonding in the novel chemical systems in which they are hosted.

**Author Contributions:** Conceptualization, project design, and project administration, P.R.V.; formal analysis and investigation, P.R.V. and A.V.; supervision, P.R.V.; writing—original draft, P.R.V. and A.V.; writing—review and editing, P.R.V., H.M.M., A.V., and K.Y. All authors have read and agreed to the published version of the manuscript.

**Funding:** This research received no external funding.

**Institutional Review Board Statement:** Not applicable.

**Informed Consent Statement:** Not applicable.

**Data Availability Statement:** This research did not report any data.

**Acknowledgments:** This work was entirely conducted using the various computation and laboratory facilities provided by the University of Tokyo and the Research Center for Computational Science of the Institute of Molecular Science (Okazaki, Japan). P.R.V. is currently affiliated with the University of the Witwatersrand (SA) and Nagoya University, Aichi 464-0814, Japan. A.V. is currently affiliated with Tokyo University of Science, Tokyo, Japan 162-8601. K.Y. is currently affiliated with Kyoto University, ESICB, Kyoto, 615-8245, Japan. H.M.M. thanks the National Research Foundation, Pretoria, South Africa, and the University of the Witwatersrand for funding.

**Conflicts of Interest:** The authors declare no conflicts of interest. The funders had absolutely no role in the design of the study; in the collection, analyses, or interpretation of data; in the writing of the manuscript; or in the decision to publish the results.

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
