# Peer review of "Definition of the Pnictogen Bond: A Perspective"

_inorganics, doi:10.3390/inorganics10100149_

Round 1

Reviewer 1 Report

The aim of the paper entitled “Definition of the Pnictogen Bond: A Perspective” (inorganics-1911369) is to define the concept of pnitogen bond which has recently become more and more popular as an interaction responsible for packing molecules in a crystal. Exploration of crystal database shows that presence of elements of Group 15 is connected with a special weak inter – and intramolecular interaction which can be called pnitogen bond. The work focuses on explaining the nature of the interaction as occurring between an electrophilic atom of Group 15 and another electronegative atom. In order to provide a complete definition of an interaction, specific cases are considered and the specific characteristics of the interaction are taken into account. The proposed definition of pnitogen bond is in line with definition of hydrogen bond, halogen bond and chalcogen bond proposed previously. For the above reasons, the manuscript fully deserves to be published in Inorganics.

Author Response

We are incredibly grateful to this reviewer for his excellent comments and recommendations to publish this article in its current form. 

Reviewer 2 Report

This perspective on the definition of the pnictogen bond is well written and clearly distinguishes pnictogen atoms as having the quality of being either a nucleophilic site or an electrophilic site.  The latter is what is considered a pnictogen bond, as the authors point out.  The perspective is very useful and timely for researchers in the field of utilizing non-covalent interactions in a variety of applications.

I have only minor comments for the authors:

1)  Even with IUPAC's definitions of hydrogen, halogen and chalcogen bonds in mind and properly cited, the authors should mention that these types of interactions fall under the umbrella of sigma-hole interactions.  See Politzer et al, PCCP, 15, 11178 (2013).  This could be included in the Introduction.

2)  With the importance of electrostatic potentials in identifying in many instances positive sites on pnicogens and other atoms, it would be useful for the authors to cite some general paper on electrostatic potentials and noncovalent interactions.  Some examples include:  Weiner et al, PNAS USA 79, 3754-3758 (1982); Gadre et al, J. Chem. Phys. 96, 5253-5260 (1992); Murray and Politzer, WIREs Comput. Mol. Sci. 7, e1326 (2017); Suresh et al, WIREs Comput. Mol. Sci. 12, e1601 (2022).

Author Response

We are incredibly grateful to this reviewer for his excellent comments and recommendations to publish this article after minor revisions. We answer the comments as follows: 

1)  Even with IUPAC's definitions of hydrogen, halogen and chalcogen bonds in mind and properly cited, the authors should mention that these types of interactions fall under the umbrella of sigma-hole interactions.  See Politzer et al, PCCP, 15, 11178 (2013).  This could be included in the Introduction

Reply: As suggested, we have added a few lines in the conclusion section of the revised ms, which are highlighted in red. Also, we have cited the paper of Politzer and coworkers. 

2)  With the importance of electrostatic potentials in identifying in many instances positive sites on pnicogens and other atoms, it would be useful for the authors to cite some general paper on electrostatic potentials and noncovalent interactions.  Some examples include:  Weiner et al, PNAS USA 79, 3754-3758 (1982); Gadre et al, J. Chem. Phys. 96, 5253-5260 (1992); Murray and Politzer, WIREs Comput. Mol. Sci. 7, e1326 (2017); Suresh et al, WIREs Comput. Mol. Sci. 12, e1601 (2022).

Reply: All these papers suggested by this reviewer are cited in the revised version of the ms. 

Reviewer 3 Report

The authors report a nice overview on the pnictogen bond. It is well written and organize. It can help the researchers to classify such peculiar weak interaction. 

The manuscript can be accepted in the present form. 

Author Response

We are incredibly grateful to this reviewer for his excellent view on the quality of the work presented in this paper and recommendations to publish this article in its current form.